# Floral Characterization of Pomegranate Genotypes to Improve Hybridization Efficiency

**DOI:** 10.3390/plants12010165

**Published:** 2022-12-30

**Authors:** Sufian Ikram, Waqar Shafqat, Sami Ur Rehman, Muhammad Ahsan Qureshi, Safeer ud Din, Salman Ikram, Muhammad Nafees, Muhammad Jafar Jaskani

**Affiliations:** 1Institute of Horticultural Sciences, University of Agriculture, Faisalabad 38000, Pakistan; 2Indian River Research and Education Center, Horticultural Sciences Department, Institute of Food and Agricultural Sciences, University of Florida, Fort Pierce, FL 34945, USA; 3Department of Forestry, College of Forest Resources, Mississippi State University, Starkville, MS 39762, USA; 4Department of Horticulture, University of Sargodha, Sargodha 40100, Pakistan; 5Department of Horticulture, Ghazi University, Dera Ghazi Khan 32200, Pakistan; 6Department of Horticultural Sciences, Faculty of Agriculture and Environmental Sciences, The Islamia University of Bahawalpur, Bahawalpur 63100, Pakistan

**Keywords:** fruit breeding, flower sex, pollination, fertilization, conventional breeding

## Abstract

Pomegranate (*Punica granatum*) has staminate (male), androgynous (hermaphrodite), and intermediate flower types. Floral characterization is difficult for breeding efficiency across many pomegranate genotypes in Pakistan, which is essential for pomegranate cultivar enhancements. The present research focused on the floral characterization and breeding efficiency of fifteen pomegranate genotypes. Flower sex ratio, floral morphological parameters, i.e., flower length, ovary width, flower notch, flower tip and stigma with style length, and fruit set percentage were examined during the experiment. In terms of sex ratio, male flowers were found to be higher among all genotypes. Due to clear differences in flower length, width, and heterostyly facilitating visual identification of the hermaphrodite flowers, genotype Ternab-2, Kandhari White, and Kandhari Red had higher fruit set (≥70%) among all cross combinations attempted. Genotype Sava had higher flower length and heterostyly of hermaphrodite flower type, but ovary width was not very distinct, leading to average crossing success (85–34%). In conclusion, single or combination of morphological characters can be used for accurate identification of hermaphrodite flowers, which can improve hybrid efficiency and fruit set after artificial cross-pollination.

## 1. Introduction

Pomegranate was among the first five cultivated crops, mentioned in the Old Testament of the Bible, The Holy Quran, ancient Greek mythology, and the Babylonian Talmud [1,2]. Pomegranate fruits are rich with polyphenols (punicalagin and ellagic acid) and antioxidants in its juice, peel, and seeds. Pomegranate is beneficial for a wide range of medical diseases, including cancer, HIV, heart disorders, hypertension, and even aging [3,4,5]. The antioxidant content of pomegranate is as high as that of any other food crop, which confers its medicinal importance [6,7].

Pomegranates have more than 500 globally distributed varieties, of which 50 are commercially cultivated [8]. Pomegranate has a medium juvenile period of 3–4 years, and a high number of seeds obtained from a single fruit makes hybridization a very desirable technique [9,10,11]. Pomegranate has unique floral biology; flower types are sometimes grouped into two (functional male, hermaphrodite) [12] and sometimes three (functional male, intermediate, hermaphrodite) categories [13,14]; which significantly vary from genotype to genotype, environmental and agronomic conditions [11,12,15]. The flower morphology of each flower type also varies significantly; the androecium and gynoecium of each flower type have a significant degree of variation [16]. In most angiosperms the male flowers only contain male organs (i.e., filament and anther) and females flowers only contain female organs (i.e., stigma, style, ovary, ovule) however, in pomegranate all flower types contain both male and female organs. The categorization of flower types depends upon the development/differentiation of sexual organs. In male flowers, the female part is almost negligible, and they never set fruit, followed by intermediate type where female organs are somewhat developed but if it sets fruit, it usually drops or is of very small size. The hermaphrodite flower type has fully developed female organs, and they produce the highest quality fruit. Although containing a well-developed androecium, intermediate flowers mostly possess degenerative ovaries that rarely set fruit [16]. Most of the hermaphrodite flowers are reported to contain heterostyly (style raised above anthers), which makes pomegranate an often cross-pollinated crop, and the presence of heterostyly also varies among genotypes [13,14].

Flower evolution is governed by the functional, developmental, environmental, genetic, and hormonal patterns of plant species. To ensure species fitness in response to environmental stimuli, angiosperms have modified their flower structure to attract more pollinators. Floral parallel trait evolution has been largely influenced by pollinator characteristics [17]. Since pomegranate has wide growing adaptability, it has evolved its floral structure to adapt to different environments and pollinators, and it greatly varies among genotypes. Aside from visual appearance (color, location, smell), floral phenology is also important in fertilization success. The stigma and stamen length ensure crossability among species, flower size, ovary width, pollen sac position and pollen tube length have influence on fruit set percentage [18,19,20].

Pomegranate has a medium juvenile period 3–4 years [11] and the high number of seeds obtained from a single fruit makes hybridization a very desirable technique [9,10,11]. Hybridization is a laborious and time-consuming process. For large-scale breeding programs, improving the cross-success percentage can reduce resources consumption. In pomegranate, to differentiate hermaphrodite flowers from other flower types, three parameters, i.e., length of flower, ovary width, and heterostyly, are usually adopted. However, the accuracy of these selection parameters does not follow a definite pattern, especially in the case of the hermaphrodite/intermediate type. Additionally, genotypes differ in floral structure, so flower selection for breeding could be deceptive. Variation among flower types of various genotypes in pomegranate has not been studied, and the efficiency of flower selection parameters for hybridization remains to be determined. This research hypothesized that for flower selection in pomegranate, a standard criterion can be developed that can improve hybridization success.

## 2. Results

### 2.1. Sex Ratio

The flower sex ratio in pomegranate genotypes varied greatly. Male flowers were found to be in the highest proportion (45.2–72.8%), followed by (18.3–38%) and (8.6–25.8%) of intermediate and hermaphrodite types, respectively (Figure 1).

### 2.2. Flower Morphology

#### 2.2.1. Length of Flower

Flower length is usually the adopted method for selecting flower types in pomegranates. Among the fifteen genotypes under study, flower length was statistically significant among different flower types (P-gender < 0.001) and genotypes (P-genotypes < 0.001) (Figure 2). In flower types, the flower length was highest in hermaphrodite flowers, followed by intermediate and male flowers, respectively (Table 1). Among genotypes the flower length differed significantly among Ternab 2, Sandhora, BW3 and Takht-i-Babri (Figure 2A) (Table 1).

The addition of sepals and petals can sometimes be misleading about the size of the androecium and gynoecium of flowers. In genotypes under study, the length from base of the flower to sepal notch for flower types (P-gender < 0.001) and genotypes (P-genotypes < 0.001) showed statistical significance. Simmilarly, the length differed significantly among genotypes Green Khushab, Dasi, Sava and Red. Among flower types, hermaphrodite flowers showed the highest value for length from base of the flower to sepal notch followed by intermediate and male flower types (Figure 2B) (Table 1).

Length from the base of the flower to the tip of sepal was statistically significant among flower types (P-gender < 0.001) and genotypes (P-genotypes < 0.001). Among genotypes Green Khushab, Kandhari Red, BW3, Sava and Takht-i-Babri differed significantly for length from the base of the flower to the tip of sepal (Figure 2C). In flower types, the flower length was highest in hermaphrodite flowers, followed by intermediate and male flowers, respectively (Table 1).

#### 2.2.2. Heterostyly

Heterostyly is used as an indication for the identification of hermaphrodite flowers. In current studies, statistical significance was recorded for flower types (P-gender < 0.001) and genotypes (P-genotypes < 0.001) (Figure 3). Briefly, in genotypes Ternab 2, Kanhati, Sandhora Khatta, Sava and Sandhora stigma+ style length differed significantly. Among flower types, the stigma+ style length was highest in hermaphrodite flowers in all genotypes followed by intermediate and male flowers (Figure 3A) (Table 1).

The stigma+ style+ stylopodium length was statistically significant among flower types (P-gender < 0.001) and genotypes (P-genotypes < 0.001) (Figure 3). Briefly, among all genotypes the stigma+ style+ stylopodium length of hermaphrodite flowers was highest, followed by Intermediate and male flower types. Among genotypes Ternab 2, Takht-i-Babri, Dasi, Sandhora Khatta, Sandhora and Sava showed significant difference for stigma+ style+ stylopodium length (Figure 3B) (Table 1).

### 2.3. Flower Width

#### Ovary Width

Ovary width is a direct indicator of the reproductive development of flowers. Most intermediate and all the male flowers have degenerative ovaries that do not or very rarely produce fruit. The ovary width was significantly different among genotypes (p-genotypes < 0.001) and different flower types (p-gender < 0.001) (Figure 4). Briefly, among genotypes Sandhora, Takht-i-Babri, Red and Sava the ovary width differed significantly (Moreover, among flower types, the ovary width was highest in hermaphrodite flowers, followed by intermediate and male flowers (Figure 4) (Table 1).

### 2.4. Cross-Pollination

To estimate the efficiency of selection criteria in genotypes, artificial cross-pollination was attempted before the flower morphology study of each genotype. Comparing the results from morphological studies, genotypes Ternab 2, Kandhari Red, and Kandhari White had marked differences in hermaphrodite and intermediate/male flower length parameters, which facilitated their visual selection, resulting in good crossing success (≥70%). In contrast, genotypes BW3, Takht-i-Babri, and Sandhora had interrelated mean values for intermediate/male and hermaphrodite flower length parameters, making their selection visually misleading, resulting in lower crossing success (≤50%).

Genotype Sava had a clear difference in length and heterostyly of intermediate/male and hermaphrodite flower types, but the ovary width of flowers was not very distinguishable, leading to average crossing success (85–34%); in genotype Sava, some of the crossed flowers must have degenerative ovary from intermediate flower types, which reduced fruit set percentage. Genotype Takht-i-Babri and BW3 had similar mean values for flower length and heterostyly between intermediate and hermaphrodite flower types, but ovary width varied significantly in these flower types, fruit set percentage also had a lower range (58–23%) and (56–31%), respectively. In these flower types, selection should have been made based on ovary width difference rather than flower length or heterostyly, as all flower type is shorter, but hermaphrodite is considerably wider than other flower types (Figure 5). In all other genotypes, the fruit set percentage followed a medium range. For improvement of the crossing success percentage, the flower morphology of each flower type in genotypes subjected to hybridization should be studied prior to crossing (Table 2).

## 3. Discussion

Andromonoecy in pomegranates promotes profuse flowering, consuming viable plant nutrients, as most functional male and intermediate flowers usually do not set fruit. In each flower type, organogenesis of both androecium and gynoecium occurs, but the degree of development varies greatly and results in the production of three flower types [12]. The key developmental difference within the ovule development of functional male and female flowers was observed in inner integument development when flower bud size was around 5–13 mm [12]. In female flowers, the inner integument after outer integument formation continues cell enlargement, increase in the number of cell layers, and growth parallel to nucellus while, in contrast with male flowers. However, integument formation was observed in part of the ovules, but the inner integument did not enlarge, and the ovules showed wilting [21].

To understand the morphological and developmental differences among genotypes, we compared the morphological parameters and observed a clear difference in the morphology of the three flower types. Flower size parameters, heterostyly parameters, and ovary width showed a spectrum of range among and within flower types. The degeneration of ovary parts in male and intermediate pomegranate flower types reduces its size. Degenerated ovaries rarely produce fruit, and when they do, the size and quality of the fruit are greatly reduced [12,16]. Heterostyly parameters greatly differ among flower types in pomegranate [13]. Heterostyly can directly influence the pollination behavior of plants; in the monoecious plant, to ensure species fitness, heterostyly is an adaptive method for cross-pollination insurance that facilitates the adaptability of species [22,23]. Flower length parameters vary between pomegranate flower types, genotype, and flower position on the branch [16]. These parameters are also affected by the nutritional status of the plant and the effects of different plant growth regulators [24].

The percentage of flower type is reported to be directly related to agronomic/environmental conditions, plant health, the age of the tree, and genotypes [25,26]. In current studies, these factors were nullified because plants were grown simultaneously and maintained in the same field conditions. There was a statistical difference in the flower sex ratio among genotypes. Hermaphrodite sex was the lowest, while the male type was in the highest numbers. Similar sex ratio patterns are found in the majority of fruit crops [27]. In all plant species, especially in monoecious plants, the character of the high number of male flowers is regularly consistent, which ensures male fitness of species, but the ratio of hermaphrodite and intermediate follows a complex pattern and depends on many factors like environment, genotypes, and nutritional status of plant [23,28,29]. The external ethephon has been reported to increase the number of bisexual flowers in pomegranate [21].

In fruit crops, cross-compatibility among genotypes is an important factor influencing fruit set and fruit quality [30,31]. Pomegranate is an often-cross pollinated crop, and most genotypes are reported to be self-compatible. However, self-compatibility varies among genotypes, and fruits produced through self-pollination have been reported to have a lower aril number and a smaller size. Contrarily, the cross pollination among genotypes showed increased aril number and fruit size [32,33]. Pollen sources have also been reported to influence fruit set, when three different genotypes were used for cross combination different ratio of fruit set was observed. However, the pollen source with the lowest fruit set produced the best quality fruit, indicting the Xenia effect [34].

Genes regulate flower development. In pomegranate, the *pgINO* gene in the *pgYABBYs* gene family is reported to regulate organ differentiation in pomegranate flower types [35]. Previously [20] reported three AG homology genes responsible for male sterility in pomegranate functional male flowers. *AGL19* and *AGL8* were up-regulated, which plays an important role in flower transition, while *AGL62* was down-regulated, promoting nucleus degeneration.

Visual selection parameters (flower size, ovary width, heterostyly) were only found to be successful in a short number of genotypes (Ternab 2, Kandhari White, Kandhari Red), proven by higher fruit set percentage in those cross combinations, while in other genotypes the degree of variation among flower types was not so apparent, hence lower fruit set percentage. To distinguish between intermediate and hermaphrodite flower types before hybridization, morphological studies should be carried out in subjected growing season, and standards should be set to improve fruit set percentage. For hermaphrodite flower selection the combination of all three selection parameters (i.e., flower length, heterostyly, flower width) cannot be applicable for every genotype. Sometimes the length is greater in both flower types (intermediate/hermaphrodite), but the ovary width varies; sometimes, heterostyly is absent in all flower types. So, for proper identification, a single or combination of factors should be utilized.

Flower size directly affects the quality and number of seeds in the fruit [12]. The perfect flower selection can successfully enhance the fruit set percentage, saving valuable resources in a breeding program. Flower thinning is a standard operation for many fruit trees [36,37,38]. In pomegranate, removing male flowers at an early stage can save vital plant nutrients while improving the quality of remaining fruits.

## 4. Materials and Methods

### 4.1. Plant Material and Growing Conditions

Fifteen screened out diverse pomegranate genotypes were selected from Pomegranate Germplasm Block. Square#32, Institute of Horticultural Sciences, University of Agriculture, Faisalabad, Pakistan (31° 45° N, 73° 13° E with average altitude of 185 m) (Figure 6) (Table 3). Plants were kept in the same block and subjected to the same agronomic practices to reduce variability caused by environmental and cultural factors. Three uniform, healthy 6–7-year-old pomegranate trees of each genotype were selected for floral morphology and cross-pollination studies. Flower morphology was studied at the mid flowering stage (mid-April) during the growing season of 2019, and subsequently, cross-pollination was also carried out during the same flowering season.

### 4.2. Flower Morphology

Freshly opened flowers from selected genotypes of pomegranate were collected in the early morning and immediately taken to the lab, where they were grouped into three categories based on their visual appearance: functional male, intermediate, and hermaphrodite. For even distribution, flowers from both lateral and terminal positions on the branch and in singular and cluster form were collected. The visual categorization of hermaphrodite flowers was based on the urn-shaped ovary, larger size, and visible heterostyly. Intermediate flowers were selected based on their slightly shorter size, medium-sized ovary, and intermediate length of stigma/style. Meanwhile, functional male flowers were selected based on the bell-shaped ovary, smaller size, and invisibility of the stigma/style. Flower selection parameters were coupled with morphological studies of each flower type and subsequent cross-pollination to determine the accuracy of these selection parameters. After counting, the flower sex percentage was calculated and flowers were dissected, and the following morphological parameters were measured using a digital Vernier caliper; (1) complete flower length (2) length from the base of flowers to sepal notch (3) length from the base of flowers to the tip of sepals (4) length of stigma+ style+ stylopodium (5) length of stigma+ style (6) ovary width.
$$\mathrm{Flower}\;\mathrm{sex}\;\left(\%\right)=\frac{\mathrm{No}.\;\mathrm{of}\;\mathrm{each}\;\mathrm{flower}\;\mathrm{type}\;\left(\mathrm{male},\;\mathrm{intermediate},\;\mathrm{hermephrodite}\right)}{\mathrm{Total}\;\mathrm{number}\;\mathrm{of}\;\mathrm{flowers}}\times 100$$

### 4.3. Cross Breeding Pollination and Fertization

Cross-pollination of selected genotypes was carried out during the entire growing season. Crosses were reciprocally arranged to determine the success of each combination. Mature, unopened flower buds at the balloon stage of male flowers (identified by smaller size) were collected after removing sepals, petals, stigma, and style. Flower buds were kept under a 100- Watt lamp overnight. The shedded pollen grains were collected and used for pollination the next morning. About to open flower buds of hermaphrodite flowers (identified by their large size and urn-shaped ovary) at the partially open stage were emasculated by removing sepals, petals, and stamens. After emasculation, the pollen grains of desired male plants collected the previous day were dusted on the stigma of emasculated flowers using a camel hairbrush. Stigma receptivity was assessed by observing a sticky stigmatic solution on the stigma; if the stigma was not receptive, it was covered with a butter paper bag and crossed the next day. After pollen dusting, the flower buds were covered with butter paper bags and tagged accordingly (Figure 7). The remaining flowers on the branch were removed to increase crossing success.

The number of fruits set in each cross combination was recorded after six weeks of pollination, and the percentage was worked out using the following formula:$$Fruit\;Set\;\left(\%\right)=\frac{Number\;of\;fruits\;developed}{Total\;number\;of\;crossed\;flowers}\times 100$$

### 4.4. Statistical Analysis

The experiment was laid out according to a randomized complete block design. The analysis of variance was used to determine statistical significance among genotypes and flower types. For post hoc analysis, Tukey’s HSD test was used to determine significance among means for at a 95% confidence interval. Basic descriptive statistic was performed to calculate the range and median of three flower types for flower morphology parameters. Flower sex percentage and fruit set percentage were calculated using the formulas stated above. The graphs were constructed using SIGMA Plot 13.0.

## 5. Conclusions

In pomegranate flower, morphology greatly differs among flower types and genotypes. Flower length, flower width, and heterostyly were used to distinguish between hermaphrodite and intermediate flower types. The morphological study provided insights into the structural difference between the androecium and gynoecium of different flower genotypes, which can be employed to identify the visual difference among hermaphrodite and intermediate flower types. Single or combinations of morphological parameters can be utilized for the identification of hermaphrodite flowers, hence improving hybridization success.

## Figures and Tables

**Figure 1 plants-12-00165-f001:**
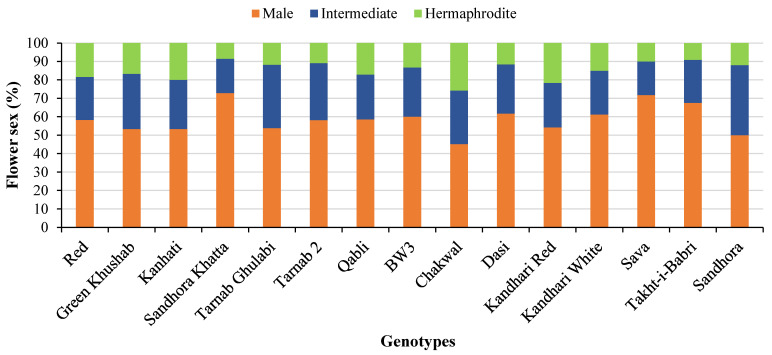
Flower sex ratio in different pomegranate genotypes. Stacked columns represent flower sex percentage, with differences in color corresponding to three flower sex types.

**Figure 2 plants-12-00165-f002:**
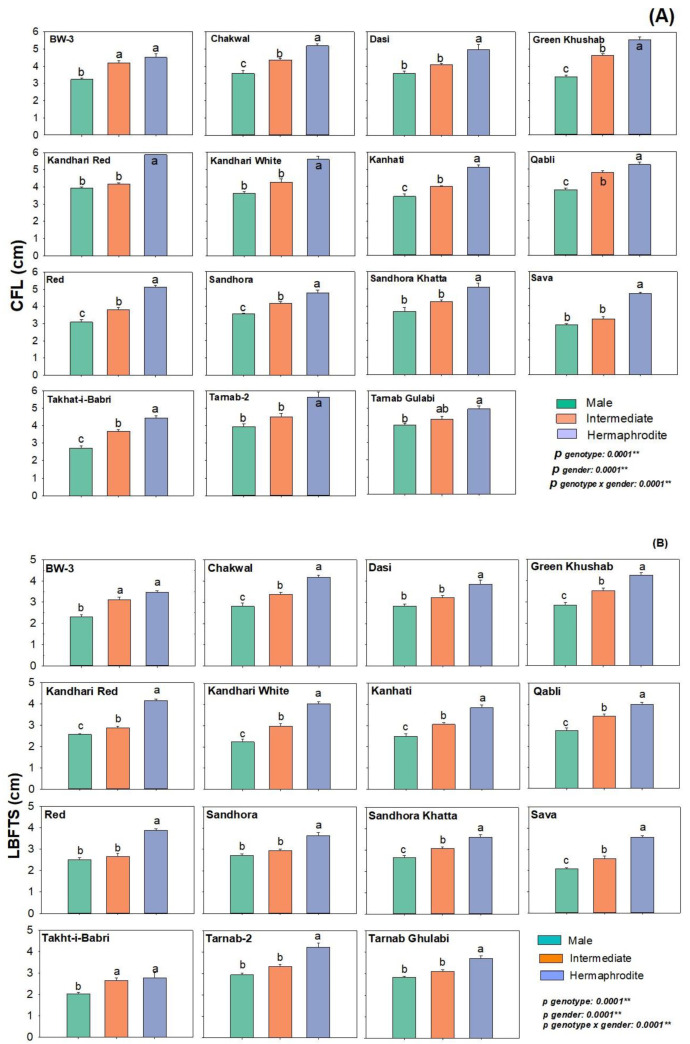
Difference in flower length parameters among three flower types in pomegranate genotypes (**A**) complete flower length (CFL) measured from base of the flower petal tip (**B**) length from base of the flower to tip of sepal (LBFTS) (**C**) length from base of the flower to sepal notch (LBFSN). After dissecting each flower type, the length was measured in centimeters (cm). The graph represents the mean ± S.E. of five replicates for each flower type. Different letters indicate statistical significance among flower types (Tukey’s HSD *p* < 0.05), ** indicates significance at *p* < 0.01.

**Figure 3 plants-12-00165-f003:**
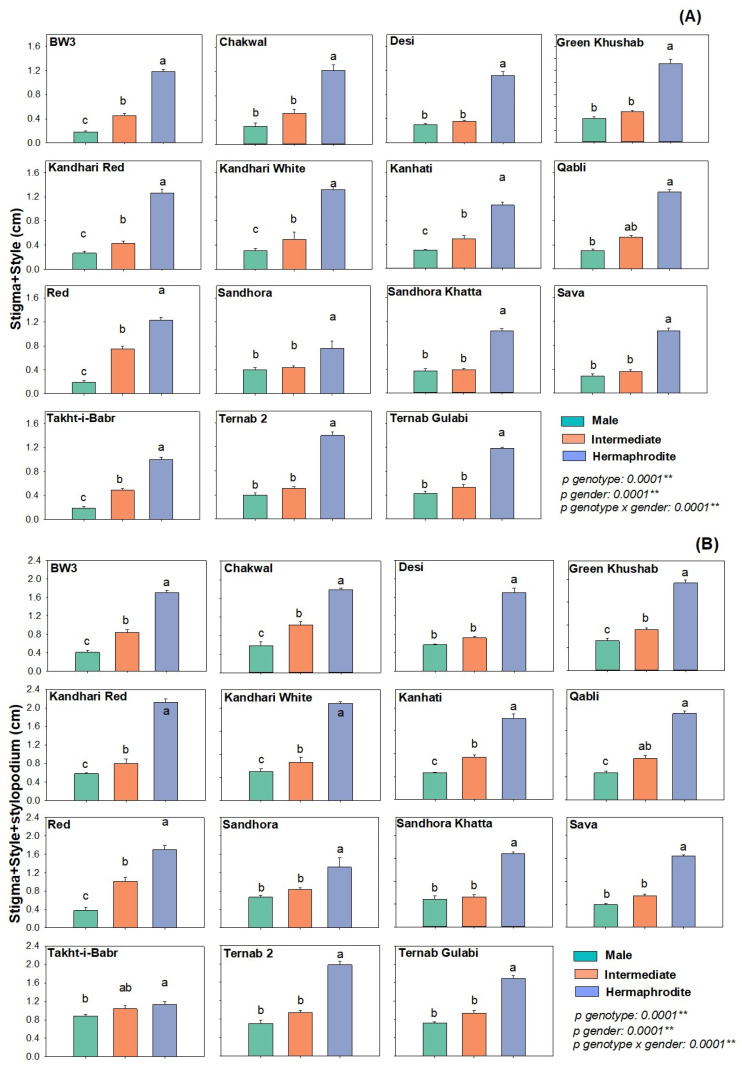
Heterostyly parameters of pomegranate genotypes in different flower types. (**A**) Stigma + Style length measured from base of stigma to top end of style (**B**) stigma+ style+ stylopodium length measured from base of stigma including style length and stylopodium top end. After dissecting each flower type, the length was measured in centimeters (cm). The graph represents the mean ± S.E. of five replicates for each flower type. Different letters indicate statistical significance among flower types (Tukey’s HSD *p* < 0.05), ** indicates significance at *p* < 0.01.

**Figure 4 plants-12-00165-f004:**
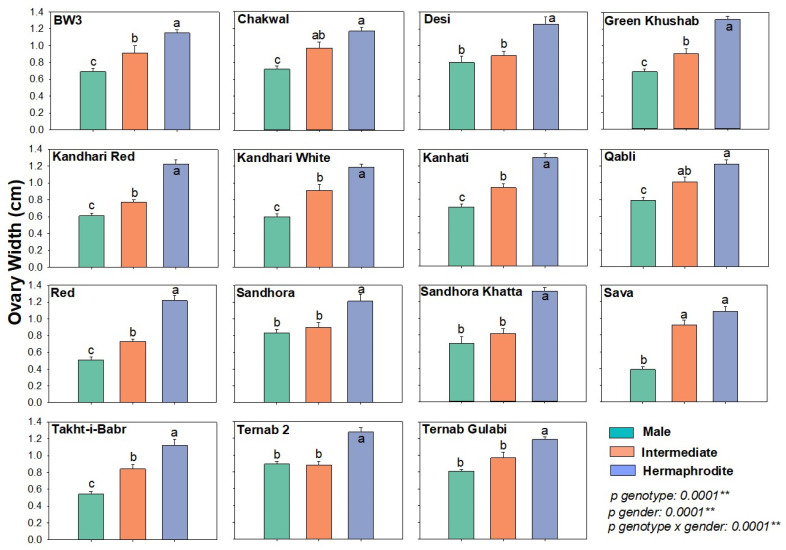
Ovary width (OW) of different flower types in pomegranate genotypes. The width of the ovary was measured in (cm) after dissection of each flower type. After dissecting each flower type, the length was measured in centimeters (cm). The graph represents the mean ± S.E. of five replicates for each flower type. Different letters indicate statistical significance among flower types (Tukey’s HSD *p* < 0.05), ** indicates significance at *p* < 0.01.

**Figure 5 plants-12-00165-f005:**
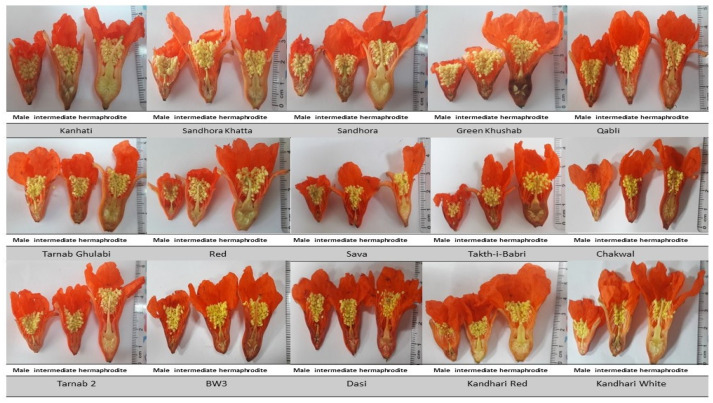
Dissection of three flower types in pomegranate genotypes. Male (**left**) intermediate (**center**) Hermaphrodite (**right**).

**Figure 6 plants-12-00165-f006:**
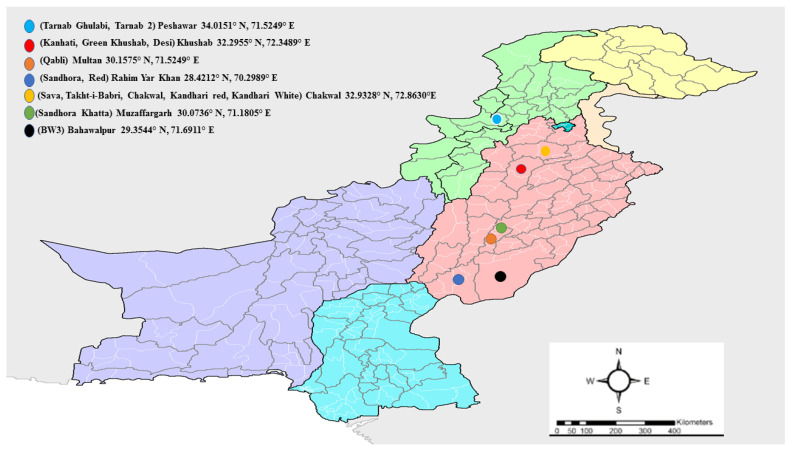
Distribution of pomegranate genotypes based on their site of collection from Pakistan. Colored dots show different collection locations on the map.

**Figure 7 plants-12-00165-f007:**
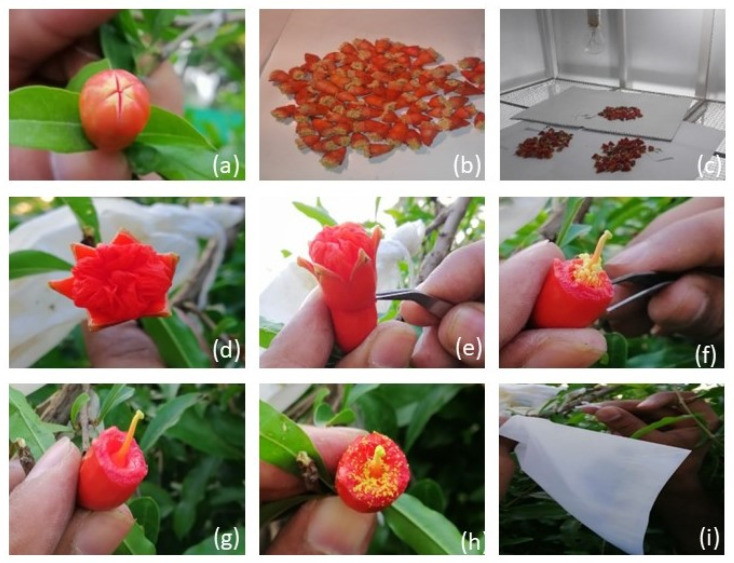
Crossing procedure in pomegranate genotypes. (**a**) bud at balloon stage (**b**) flower buds after removal of sepals, petals, stigma, and style (**c**) flower buds kept under lamp for dehiscence of pollens (**d**) flower bud at close petal stage (**e**,**f**,**g**) emasculation (**h**) pollen dusting (**i**) bagging and tagging.

**Table 1 plants-12-00165-t001:** Floral morphological parameters of pomegranate genotypes.

Parameters (cm)	Flower Types					
	Male		Intermediate		Hermaphrodite	
	Range	Median	Range	Median	Range	Median
Complete flower length	2.72–3.94	3.58	3.24–4.82	4.18	4.44–5.88	5.12
Length from the base of flowers to sepal notch	1.24–2.02	1.78	1.42–2.39	2.06	2.25–3.06	2.83
Length from the base of flowers to the tip of sepals	2.04–2.94	2.64	2.56–3.54	3.08	2.77–4.26	3.86
Length of stigma+ style+ stylopodium	0.38–0.72	0.59	0.66–1.04	0.9	1.13–2.12	1.7
Length of stigma+ style	0.19–0.43	0.31	0.36–0.75	0.5	0.76–1.39	1.18
Ovary width	0.39–0.89	0.7	0.73–1.01	0.91	1.08–1.33	1.22

The range is from the minimum to maximum value of each flower type in all genotypes, while the median shows the mid-point value for each flower type.

**Table 2 plants-12-00165-t002:** Fruit set percentage among different cross-combinations in pomegranate genotypes (female × male).

Cross Combination (Female × Male)	Fruit Set (%)	Cross Combination (Female × Male)	Fruit Set (%)
Kandhari White × Chakwal	95.6	BW3 × Tarnab Ghulabi	58.3
Tarnab Ghulabi × Tarnab 2	92.8	Sandhora × Sandhora Khatta	56.6
Ternab 2 × Red	91.8	Takht-i-Babri × Green Khushab	56.3
Ternab 2 × Kandhari White	89.3	Red × BW3	51.2
Ternab 2 × Sava	87.2	Sandhora Khatta × BW3	47.8
Kandhari White × Tarnab Ghulabi	85.7	Red × Takht-i-Babri	46.6
Ternab 2 × Takht-i-Babri	85.4	Tarnab Ghulabi × BW3	45.7
Kandhari White × Kandhari Red	84.3	Sandhora × Ternab 2	45.7
Kandhari White × Sava	84.1	Qabli × Takht-i-Babri	45.5
Kandhari White × Qabli	82.5	BW3 × Red	43.8
Kandhari Red × Qabli	82.4	Takht-i-Babri × Kandhari White	41.3
Sandhora Khatta × Sava	82.2	Takht-i-Babri × Sava	41.2
Tarnab Ghulabi × Sava	81.7	Takht-i-Babri × Kandhari Red	36.4
Ternab 2 × Sandhora	79.8	Sava × Sandhora	34.6
Kandhari White × Ternab 2	77.4	Red × Kandhari Red	32.1
Kandhari White × Ternab Ghulabi	75.6	BW3 × Kandhari White	32.1
Green Khushab × Takht-i-Babri	74.5	Takht-i-Babri × Sandhora	31.3
Qabli × Kandhari White	67.5	BW3 × Kandhari Red	29.4
Qabli × Khatta Khushab	64.3	Red × Kandhari White	28.7
Qabli × Kandhari Red	59.5	BW3 × Ternab 2	23.6

**Table 3 plants-12-00165-t003:** Characteristics of pomegranate germplasm used for experiment [39,40].

Genotype Name	TSS (Brix)	TA (g/L)	Fruit Rind Color	Rind Thickness (mm)	Aril Color	WPI Seed Hardiness (%)	Seed Area (mm^2^)	Fruit Weight (g)
Tarnab Ghulabi	14.2	0.91	Pinkish red	4.47	Red	3.99	7.89	±289
Kanhati	13.2	0.15	Yellowish green	4.09	Pinkish	6.11	10.87	±70
Tarnab 2	N/A							
Qabli	16.8	0.44	Reddish-green	3.34	Pinkish-white	4.89	9.87	±220
Green Khushab	13.2	0.15	Pinkish-green	3.99	Pinkish-white	3.22	9.17	±170
Sandhora	12.9	0.11	Pinkish-green	3.77	Pinkish-red	6.78	9.91	±221
Sandhora Khatta	11.2	0.25	Pinkish-yellow	3.66	Pinkish-white	4.78	8.96	±184
Sava	15.9	0.09	Yellowish	3.22	White-pink	3.89	11.65	±278
Takht-i-Babri	13	0.23	Reddish	2.48	Pinkish-white	13.65	8.97	±46
Chakwal	13.8	0.15	Reddish	2.01	white	15.67	9.22	±40
Kandhari red	14.9	0.17	Red	2.87	Red	3.59	10.78	±240
Kandhari White	15.1	1.4	Pinkish-red	2.99	Red	5.33	8.98	±220
Dasi	15.7	0.12	Reddish-pink	2.24	Pinkish-white	5.78	9.87	±118
Red	14.5	0.15		3.77		4.67	9.78	±167
BW3	N/A							

Abbreviations: TSS; total soluble solids, TA: titratable acidity, WPI; woody portion index, Data represents mean values for fruit characteristics of pomegranate genotypes reported in previous studies.

## Data Availability

The data that support the findings of this study are contained within the article and are available from the corresponding author upon reasonable request.

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
