# Peer review of "Floral Characterization of Pomegranate Genotypes to Improve Hybridization Efficiency"

_plants, 2022, doi:10.3390/plants12010165_

Round 1

Reviewer 1 Report

The revised manuscript is devoted to comparison of selected quantitative characteristics between different floral types in a set of pomegranate cultivars.
First, the results of this work are very practically oriented. I am sure they would better suit any journal of agricultural scope (e.g. Horticulture of the MDPI).
Second, I have problems understanding the key goal of this survey. If the idea was to work out an algorithm of distinguishing between floral types, it seems an excessive work as authors differentiated collected flowers into three types before any measurements. It suggests as if this differentiation is possible without making numerous measurements.
Third, I disagree with the authors' idea that lower fruit set is because of difficulty to distinguish between floral types when performing crosses. It is a possible reason but not a single one. Why not to suggest that different genotypes possess unequal cross-compatibility? To my surprise, authors did not study seed set which may be also indicative.
Fourth, this manuscript needs very serious elaboration before beeing resubmitted. Its language requires a deep revision with assistance of a native speaker. In many cases, I did not manage to understand what authors meant. This manuscript is not formatted according to Plants' guide for authors. The statistical part of work is very obscure. All graphs need to be adjusted and supplied with clear captions. I cannot understand if differences between floral types are statistically significant or not, as authors compare them in absolute values (e.g. <0.5 cm). I have left numerous comments and suggestions considering both language and style, but much more work is needed.
To sum up, I don't think this paper can be accepted for publication in Plants in its present state.

Author Response

Response to review 1

>Comment

Second, I have problems understanding the key goal of this survey. If the idea was to work out an algorithm of distinguishing between floral types, it seems an excessive work as authors differentiated collected flowers into three types before any measurements. It suggests as if this differentiation is possible without making numerous measurements.

>Reply

For practical pomegranate breeding there is this preconceived idea that if flower has wider ovary, large flower length and style above anther. It is most suitable for hybridization and breeders usually apply this throughout the spectrum. Here we applied the same criteria for flower selection but coupled with morphological studies to determine accuracy of theses selection parameters. Question was, can these parameters be applied throughout the genotypes as different genotypes has different structural characteristics among flower types. Flower morphological studies identified differences among androecium and gynoecium structure of three flower types in pomegranate genotypes.

>Comment

Third, I disagree with the authors' idea that lower fruit set is because of difficulty to distinguish between floral types when performing crosses. It is a possible reason but not a single one. Why not to suggest that different genotypes possess unequal cross-compatibility? To my surprise, authors did not study seed set which may be also indicative.

>Reply

Cross-compatibility can also be a reason for lower fruit set, however in pomegranate there have been no reports for cultivar incompatibility. As pomegranate is often self-pollinated crop artificial cross pollination has only been reported to increase fruit set percentage and improve quality. Literature reports that hermaphrodite flowers tend to produce fruits with larger size and better quality. Which is also influenced by either fruit is self or cross pollinated. This experiment focused on studying flower morphology in relation with accurate flower selection to increase hybridization success. The fruit size could have been an indicator but it could have also been due to just the fact that the flowers are cross pollinated. That’s why fruit quality parameters and seed set were not recorded.

>Comment

Fourth, this manuscript needs very serious elaboration before being resubmitted. Its language requires a deep revision with assistance of a native speaker. In many cases, I did not manage to understand what authors meant. This manuscript is not formatted according to Plants' guide for authors. The statistical part of work is very obscure. All graphs need to be adjusted and supplied with clear captions. I cannot understand if differences between floral types are statistically significant or not, as authors compare them in absolute values (e.g. <0.5 cm). I have left numerous comments and suggestions considering both language and style, but much more work is needed.

>Reply

The corrections are made in the manuscript as per the instructions.

Reviewer 2 Report

Pomegranates are characterized by the presence of two or three functional categories of flowers, which show morphological variation and occur in different proportions in different genotypes. This fact is of great importance for the hybridization technique.

The authors of the study examined the variation among flower types of 15 pomegranate genotypes according to established criteria (6 morphological parameters). As suggested by the authors, this selection of pomegranate flowers may contribute to improvement of hybridization success.

The manuscript presents an interesting and valuable study, especially in practical terms.

Comments

Fig.1 -  The description does not provide information about the country of origin of the map.

Table 2 - there are no letters showing significance.

-  What is the place of work of the last author? What does "7" mean?

Author Response

Response to review 2

>Comment

Fig.1 - The description does not provide information about the country of origin of the map.

>Reply

Corrections are made in the manuscript

>Comment

Table 2 - there are no letters showing significance.

>Reply

Corrections are made in the manuscript

>Comment

What is the place of work of the last author? What does "7"mean?

>Reply

 Corrections are made in the manuscript

Round 2

Reviewer 1 Report

As I may see, authors have made a serious work on their manuscript to improve it. I am especially glad graphs are now clearer and easier to read.
However, language and style still need improvement. There are numerous points which are probably printing errors (e.g. 'lattering' instead of 'lettering').
All questions which I asked could be also asked by any other reader. It is therefore meaningful not only to reply to my comments but also incorporate some clarifications into the text. I recommend to add some notes about cross-compatibility to the Discussion section.
The results of statistical work need to be described carefully and correctly. Statements like 'The male and intermediate flower type in genotypes… were also statistically non-significant' are not correct. Only hypotheses can be significant or not. In this case, it is probably more correct that values of some characteristics (not floral types!) differed significantly between certain genotypes. The whole text should be checked for consistency in such points.
I still cannot understand the Figure 6. Both axes are labeled as 'Fruit set (%)'. It is unclear what is X and what is Y and why they are different. If both axes represent the same values, the scatterplot would become a line. It needs to be explained.
To sum up, this paper needs a second round of a serious elaboration before being accepted for publication.  

Author Response

Dear Reviewer,

Thanks for the review my Manuscript. We have improved our manuscript in the light of your valuable comments. Please take a look on attached manuscript and replied in following bullets.

  • Language and style still need improvement.

Reply: Changes have been made in the manuscript  

  • There are numerous points which are probably printing errors (e.g. 'lattering' instead of 'lettering').
    All questions which I asked could be also asked by any other reader. It is therefore meaningful not only to reply to my comments but also incorporate some clarifications into the text.

Reply: Changes have been made in the manuscript 

  • I recommend to add some notes about cross-compatibility to the Discussion section.

Reply: Changes have been made in the manuscript 

  • The results of statistical work need to be described carefully and correctly. Statements like 'The male and intermediate flower type in genotypes… were also statistically non-significant' are not correct. Only hypotheses can be significant or not. In this case, it is probably more correct that values of some characteristics (not floral types!) differed significantly between certain genotypes. The whole text should be checked for consistency in such points.

Reply: Changes have been made in the manuscript 

  • I still cannot understand the Figure 6. Both axes are labeled as 'Fruit set (%)'. It is unclear what is X and what is Y and why they are different. If both axes represent the same values, the scatterplot would become a line. It needs to be explained.

Reply: Changes have been made in the manuscript

Thanks for your precious time and suggestions. I really appreciate your feedback.

Best

Waqar Shafqat

Round 3

Reviewer 1 Report

Before accepting this paper for publication, two things need to be done.
1. Please elaborate Figures 2-4 and their captions. At the moment, you write 'p-value for genotypes and flower type represents significance'. There can be no 'significance' for anything. For example, correlation can be significant or not; differences between two groups can be significant or not. Which hypotheses did you test? As I may suggest, letters (a, b, c) indicate differences between different floral types. However, this needs to be rephrased like this: 'represents statistical significance among flower types' --> 'represent statistically significant differences between flower types (groups labeled with the same letter differ insignificantly)'.
However, it is unclear what you mean by 'p genotype 0.0001*'. Please elaborate this to make clearer what you tested and what the results were.
2. As I may see, you have changed a graph in Figure 6. I think it is not readable and recommend you to change it again. I think the best is to represent fruit sets as a semi-square matrix with maternal genotypes as columns, paternal genotypes as rows (or vice versa) and percentage of fruit set in each cell. It can be also vivid to color each cell with colors like those you applied in your former Figure 6 with range between blue and yellow corresponding to different frequencies.
After making these changes, this paper can be recommended for publication in Plants.

Author Response

Dear Reviewer,

We have improved our manuscript according to your suggested comments and replied to them in following comments

>Comment

Please elaborate Figures 2-4 and their captions. At the moment, you write 'p-value for genotypes and flower type represents significance'. There can be no 'significance' for anything. For example, correlation can be significant or not; differences between two groups can be significant or not. Which hypotheses did you test? As I may suggest, letters (a, b, c) indicate differences between different floral types. However, this needs to be rephrased like this: 'represents statistical significance among flower types' -->'represent statistically significant differences between flower types (groups labeled with the same letter differ insignificantly)'. However, it is unclear what you mean by 'p genotype 0.0001*'.Please elaborate this to make clearer what you tested and what the results were.

>Reply

I elaborate the figures according to your suggestion. I analyzed my research data with two factor factorials with one factor as a genotype, second in gender and then interaction of these two factors. I put lettering just for gender to understand the significance of each genotype in sense of flower type.  I tried to improve caption with statistically analysis to clear reader understanding.

>Comment

As I may see, you have changed a graph in Figure 6. I think it is not readable and recommend you to change it again. I think the best is to represent fruit sets as a semi-square matrix with maternal genotypes as columns, paternal genotypes as rows (or vice versa) and percentage of fruit set in each cell. It can be also vivid to color each cell with colors like those you applied in your former Figure 6with range between blue and yellow corresponding to different frequencies.

>Reply

I changed the figure to table format with name of cross and fruit set %. Now it looks clear for reader. I tried to make different figures, but it was confuse looking according to reader understanding.
